# Surgical Outcomes, Long-Term Recurrence Rate, and Resource Utilization in a Prospective Cohort of 165 Patients Treated by Transanal Total Mesorectal Excision for Distal Rectal Cancer

**DOI:** 10.3390/cancers15041190

**Published:** 2023-02-13

**Authors:** Severin Gloor, Gioia Pozza, Rebekka Troller, Markus Wehrli, Michel Adamina

**Affiliations:** 1Klinik für Viszeral- und Thoraxchirurgie, Kantonsspital Winterthur, 8401 Winterthur, Switzerland; 2Department of Visceral Surgery and Medicine, Inselspital, University of Bern, 3010 Bern, Switzerland; 3Faculty of Medicine, University of Basel, 4056 Basel, Switzerland

**Keywords:** rectal cancer, laparoscopic surgery, transanal total mesorectal excision, taTME, total mesorectal excision, local recurrence

## Abstract

**Simple Summary:**

A transanal total mesorectal excision (taTME) is a smart alternative to a conventional TME. Consecutive patients with distal rectal cancer treated by a taTME were prospectively analyzed. Median values were reported as outcomes measures. One hundred sixty-five patients (67% male and 33% female) with a tumor 7 cm from the anal verge were followed for 50 months. The resection margins were threatened in 25% of the patients, while 75% of the patients received neoadjuvant radiochemotherapy. A good mesorectal dissection and clear margins were achieved in 96% of the specimens, and 27 lymph nodes were harvested. Ninety-day major morbidity affected 36 patients (21.8%), including 12 with anastomotic leakages (7.2%). A recurrence occurred locally in 9 patients (5.4%), and 44 patients had distant metastasis (26.7%). The five-year disease-free survival and overall survival were 67% and 90%, respectively. A two-team taTME saved 102 min of operative time and EUR 1385 when compared to a one-team approach. Transanal total mesorectal excision produced sound surgical quality and excellent oncologic outcomes.

**Abstract:**

A transanal total mesorectal excision (taTME) is a smart alternative to a conventional TME. However, worrisome reports of a high recurrence and complications triggered a moratorium in a few countries. This study assessed the outcomes and resource utilization of a taTME. Consecutive patients with distal rectal cancer treated by a taTME were prospectively included. Outcomes were reported as the median and interquartile range (IQR). One hundred sixty-five patients (67% male and 33% female) with a tumor 7 cm (IQR 5–10) from the anal verge were followed for 50 months (IQR 32–79). The resection margins were threatened in 25% of the patients, while 75% of the patients received neoadjuvant radiochemotherapy. A good mesorectal dissection and clear margins were achieved in 96% of the specimens, and 27 lymph nodes (IQR 20–38) were harvested. Ninety-day major morbidity affected 36 patients (21.8%), including 12 with anastomotic leakages (7.2%). A recurrence occurred locally in 9 patients (5.4%), and 44 patients had a distant metastasis (26.7%). The five-year disease-free survival and overall survival were 67% and 90%, respectively. A multivariate analysis found a long operation and frailty predicted an anastomotic leak, while a positive distal margin and lymph nodes predicted a local recurrence and distant metastasis. A two-team taTME saved 102 min of operative time and EUR 1385 when compared to a one-team approach. Transanal total mesorectal excision produced sound surgical quality and excellent oncologic outcomes.

## 1. Introduction

While cancer is on the rise worldwide, and Europe counts 500,000 new colorectal cancer yearly, the survival and quality of life of patients with rectal cancer have markedly improved over the past decades [1,2]. The major achievements were the promotion of a total mesorectal excision (TME) in 1982 [3] and of neoadjuvant radiochemotherapy, which has led to a reproducible reduction in local recurrence rates from 16% to less than 5% and improved disease-free and overall survival. Since then, the widespread adoption of minimally invasive surgery has reduced surgical trauma and improved patients’ recovery and clinical outcomes. Nevertheless, the anatomical limitations of a narrow pelvis, fatty mesorectum, and bulky distal tumor continue to pose a challenge even to the most experienced colorectal surgeon. Moreover, neoadjuvant treatments add to the anatomical challenges by altering tissue quality and handling [4,5].

However, while minimally invasive approaches provide unquestioned benefits regarding recovery and quality of life, a laparoscopic TME remains contested. Indeed, large randomized controlled trials failed to demonstrate oncologic superiority [6,7], while a conventional laparoscopic approach was associated with a large amount of residual mesorectum and an incomplete TME [8]. Robotic surgery was proposed as a technical refinement in this context. Unfortunately, both a large randomized controlled trial and a meta-analysis of cohort series failed to show a substantial advantage in clinical outcomes when a robotic TME was compared to a conventional laparoscopic TME surgery [9,10,11]. The requirement of multiple stapler firings is a further issue for both laparoscopic and robotic TMEs, as the anastomotic leak rate increases with every additional cartridge fired [12]. Finally, the inability to visualize the rectal lumen and lower cancer edge when dissecting the mesorectum may lead to inadequately long distal margins and worse function.

A transanal TME (taTME) was proposed in 2009 as an innovative approach with many purported benefits claimed in early publications [13,14]. Indeed, performing a bottom-up TME allowed for a tailored oncologic distal margin, a safer single-stapling technique, a facilitated excision of the lower rectum, and an improved visualization of the pelvic nerves to be preserved [8]. Such claims should translate into better clinical outcomes, which are currently being assessed by ongoing large multi-center randomized controlled trials comparing a laparosopic TME to a taTME [15,16]. As with any new technique, a learning curve must be mastered through proper training [17], in particular to avoid previously unknown pitfalls—among them injury to the prostatic urethra and CO_2_ embolism [18,19]. The present institution belongs to the early adopters of a taTME, with the first procedure being performed in 2013, participation in the International taTME Registry since 2014 [20,21], and leadership in professional education with the publication of the St. Gallen consensus on safe implementation of a taTME in 2018 [22] and of an international expert consensus guidance on indications, implementation, and quality measures for a taTME in 2020 [23]. These consensuses were endorsed by the European Association of Endoscopic Surgery and by the European Society of Coloproctology. 

Ideally, a taTME is performed in a two-team technique with two surgeons operating simultaneously transabdominally and transanally [22,23]. Indeed, a difficult pelvic dissection is best performed when carried out concurrently from both sides. In an era where healthcare resources are scarce, and their utilization is scrutinized, the additional expenses required to perform a two-team procedure must be justified by additional benefits. Moreover, the safety of a taTME has recently been questioned with reports of an early, multifocal pelvic recurrence and a high anastomotic leak rate.

This prospective study aims to assess the surgical and oncologic outcomes of a taTME, including an assessment of resource utilization and benefits when performing a one-team or two-team taTME.

## 2. Materials and Methods

All patients with a distal rectum cancer referred to the Kantonsspital Winterthur were considered for inclusion in the present cohort series and were prospectively included in an audited clinical database of the European Cancer Centre Certification program. A further independent audit was performed by the Swiss Conference of the Cantonal Ministers of Public Health to ensure process compliance and high outcome quality. Finally, the present colorectal surgery unit participates in the Enhanced Recovery After Surgery (ERAS) program and audit [20,24]. 

Patients eligible for this study were adults with distal rectal cancer within 10cm from the anal verge, measured by rigid proctoscopy prior to initiation of neoadjuvant therapy, and who qualified for a minimally invasive curative resection. Over the full duration of the study, less than 5% of rectum cancer patients were operated by a non-minimally invasive restorative approach. Exclusion criteria were benign diseases, non-curative resection, abdominoperineal resection, Hartmann’s resection, multivisceral resection, pelvic exenteration, non-elective resection, and objection to subsequent use of personal health data.

The present single-center prospective cohort analysis was approved by the Cantonal Ethical Committee (BASEC Nr. 201902485), and the study protocol was registered (ClinicalTrials.gov Identifier NCT05112016). 

### 2.1. Statistical Analysis

Categorial data were presented as number of patients and percentages. Continuous data were expressed as median and interquartile range (IQR). Length of stay, reintervention, number, and grade of 30-day and 90-day complications were reported according to the Clavien–Dindo classification [25]. Anastomotic leakage was defined as a defect of the intestinal wall at the anastomotic site, resulting in communication between the intra- and extraluminal compartments [26]. Major morbidity was defined as any complication ≥3a grade. Time to tumor recurrence was defined as the first tumor relapse following curative taTME. Recurrence rates were calculated from the date of resection to the date of recurrence (local or distant recurrence) using Kaplan–Meier methodology. Disease-free survival (DFS) and overall survival (OS) were analyzed. Patients who died from other causes or were still alive at the date of the last follow-up examination were treated as censored in this study. Patients were further stratified in ‘one-team’ and ‘two-team’ cohorts. Comparisons between groups were performed with the chi-square or Fisher’s exact test for categorical variables and the Mann–Whitney U test for non-continuous variables. Risk factors for cancer recurrence and anastomotic leak were identified in univariate analysis, and those with a *p* value < 0.10 were entered into Cox multivariate regression analysis.

The surgical cost for each patient was evaluated according to hospital ledger. Relative costs of one-team vs. two-team approaches were calculated based on the pricing of the operation room time used, staff, and material costs. Costs in Swiss Francs were converted into Euros (exchange rates from 2 October 2022: https://secure.ubs.com/global/en/quotes.html).

*p* < 0.05 was considered statistically significant. Statistical analysis was performed using SPSS version^®^ 25 (IBM, Armonk, New York, NY, USA).

### 2.2. Perioperative Management

Perioperative management was standardized and in agreement with the guidelines of the European Cancer Centre Certification and the ERAS society. Clinical examination, measurement of carcinoembryonic antigen; total colonoscopy; rigid rectoscopy; CT scan of the chest, abdomen and pelvis; and pelvic MRI were performed in all patients. Whenever suspicious liver lesions were seen, a liver MRI was performed. Patients were presented at the weekly interdisciplinary tumor board for digestive cancer following completion of staging. Long-course neoadjuvant chemoradiotherapy (CRT: 50.4 Gy with concurrent capecitabine) was indicated for tumors with ≥cT3 and/or N+ on MRI or tumors with threatened circumferential (≤1 mm) or distal resection margins [27]. Restaging with pelvic MRI and rigid rectoscopy was performed 6–8 weeks following completion of neoadjuvant CRT. A taTME was performed 8–12 weeks after neoadjuvant CRT. Adjuvant chemotherapy was discussed at the postoperative tumor board and was started 6-8 weeks postoperatively, when indicated. Oncologic surveillance followed international guidelines [27]. 

### 2.3. Surgery

All patients underwent bowel preparation without oral antibiotics. Preoperative parenteral antibiotic prophylaxis with cefuroxime and metronidazol was completed 30 min before skin incision and repeated quid 4 h until completion of the procedure. Thromboembolic prophylaxis with low-molecular weight heparin was started 6 h postoperatively. 

The choice of a one-team or two-team approach was dictated by staff availability, with a two-team approach being the preferred option. A two-team approach was defined by the concurrent performance of the pelvic dissection by a board-certified digestive surgeon, who operated transabdominally, while another board-certified digestive surgeon trained in taTME operated transanally. Once pelvic dissection was completed, surgery was completed by a single board-certified digestive surgeon.

Surgical steps were standardized, and all procedures were performed in presence of one fully trained taTME surgeon [22,23]. Laparoscopic abdominal steps included high ligation of the inferior mesenteric artery and of the vein at the pancreas lower border, routine full splenic flexure mobilization, and pelvic dissection along the TME plane. Transanal dissection took advantage of a 3D laparoscope (Endoeye 3D, 10 mm scope, Olympus Europa, Hamburg, Germany), a LoneStar retractor (Cooper Surgical, Trumbull, CT, USA), a 3-trocar GelPoint path (Applied medical, Rancho Santa Margarita, CA, USA) and an AirSeal (Conmed, Largo, FL, USA) insufflation/evacuation system. Rectal washout with povidone iodine was performed, and then a purse string was placed to close the rectal lumen below the tumor. Following completion of the TME, pelvic washout with povidone iodine was performed. The specimen was extracted through a short suprapubic Pfannenstiel laparotomy using a wound protector. Side-to-end colorectal anastomosis was performed using a 33 mm circular stapler (EEA 4.8mm staples, Medtronic, Minneapolis, MN, USA). The anastomosis was visually inspected for integrity and adequate perfusion, and anastomotic leak testing was performed with air and saline. A loop ileostomy was routinely fashioned. 

### 2.4. Histological Evaluation

The macroscopic resection quality was graded according to Quirke [28]: A grade 3 complete taTME specimen is defined by an intact mesorectum with only minor defects smaller than 5 mm; a nearly complete grade 2 specimen harbors larger defect of the mesorectum, while not affecting the muscularis propria; an incomplete grade 1 taTME specimen reveals the muscularis propria or more in defect of the mesorectal fascia. Distal and circumferential margin were measured, and R1 resection was defined as the microscopic presence of tumor cells within 1 mm from the dissection/transection line. Pathology staging was performed according to the current TNM classification, including lymphovascular and perineural invasion. The tumor regression grade was assessed according to Dworak [29] in patients who underwent neoadjuvant CRT: TRG 0 means no regression; TRG 1 means with dominant tumor mass, fibrosis, and/or vasculopathy; TRG 2 reveals dominantly fibrotic changes with few, easy to find tumor cells; TRG 3 describes only very few tumor cells, which are difficult to find within fibrotic tissue with/without mucus; and TRG 4 means total regression with only fibrotic tissue without tumor cells. High-quality TME was defined as a complete mesorectal excision, a negative circumferential radial margin (≥1 mm), and a clear distal resection margin (≥1 mm) [30,31].

### 2.5. Economic Evaluation

Compensation for inpatient care takes advantage of the diagnosis-related group (DRG) system, which has been enforced in Switzerland since 2012. Compensation followed classification in a given DRG, compounded by a cost weight particular to each institution. The tariff structure and compensation calculation are overseen at the national level and are reviewed annually. Internal cost calculation took into account true cost that occurred when providing healthcare services. For example, one minute spent in operation theatre was internally billed CHF 12.92/EUR 13.18. This included all OR and anesthesiology staff involved in a procedure, with no difference made between one-team and two-team approaches. Conversely, qualifications and number of physicians involved were internally billed according to actual service provided. Hence, a two-team taTME generated more costs than a one-team approach because two more physicians were involved for a maximum of 3 h.

## 3. Results

### 3.1. Demographics

This prospective single-center study included 165 consecutive patients with a distal rectum cancer operated by a taTME. No patients declined to participate, yet one consented patient died of a nosocomal pneumonia during neoadjuvant CRT, so they were excluded. The primary data collection occurred between January 2014 and July 2021, with follow up updated until 3 January 2023. The Kantonsspital Winterthur is a teaching hospital affiliated with the University of Zürich that has an audited surgical case volume of more than 50 rectal cancers yearly. Specialist surgeons in training operate under the supervision of a certified digestive surgeon. The department of surgery has two trained taTME surgeons, who were involved in all the cases. The routine approach for proximal rectal cancer was a laparoscopic TME, whereas distal rectal cancer was routinely operated on by a taTME.

A total of 110 males (67%) and 55 females (33%) with distal rectal cancer, with a median age of 64 years (interquartile range (IQR) 55–76) and BMI of 26 (IQR 23–29), were analyzed. Nearly half (47%) of the patients had an American Society of Anesthesiology score of 3–4. The lower edge of the tumor was measured at a median of 7 cm (IQR 5–10) from the anal verge by rigid rectoscopy at the time of diagnosis. Pelvic MRI revealed a threatened circumferential margin in 42 patients (25%). A neoadjuvant CRT was performed in 124 patients (75%). There were no differences in the demographics, cancer stage, and neoadjuvant treatment between the one-team and two-team patients, besides a slightly younger age for the two-team patients, as shown in Table 1. 

### 3.2. Surgical Outcomes

The median operative time was 348 min (IQR 293–425), and only a single patient required a conversion to laparotomy (0.6%). The operative time was significantly shorter when patients were operated on with a two-team procedure (422 min vs. 320 min, *p* < 0.001). Three CO_2_ embolisms (1.8%) with hemodynamic instability were observed. In these, a taTME could be resumed and completed successfully following fluid resuscitation, hemostasis, and temporary interruption of the pneumopelvis for less than 10 min. There were no urethral injuries and no leakage through the purse-string suture. However, six purse strings (3.6%) were restitched immediately after primary completion. The median length of stay was 9 days (IQR 7–13, range 4 to 37 days), followed by routine discharge to a rehabilitation clinic. All-but-one patient had a diverting ileostomy, which was reversed after a median of 2 months (IQR 2–5) in 86% of the patients.

Overall, there were 91 complications prospectively reported, for a 90-day morbidity rate of 53.9%. Of these, 55 patients (33.3%) suffered from minor complications and 36 (21.8%) suffered from major complications, which required radiological, endoscopic, or surgical intervention within 3 months of surgery, including 3 deaths (2.1%). Two patients died of septic complications related to an anastomotic leak within 30 days of surgery (1.2%), and one patient died out of hospital of a myocardial infarction shortly before ileostomy reversal. Anastomotic leakage affected 12 patients (7.2%). The leaks were treated by transanastomotic vacuum therapy and laparoscopic lavage and drainage, whereas an ileostomy was fashioned in the single patient who initially refused a diverting ileostomy. All leaking anastomosis were salvaged, and the ileostomy was reversed within 6 months, except for the two patients who suffered a fatal course. A total of 14 patients (8.5%) required a reoperation within 90 days: 12 for anastomosis salvage as indicated above and 2 for the revision of a dysfunctioning ileostomy. Of note, the complication rate and complexity were higher in patients operated on by a single team. In particular, significantly more intraoperative bleeding and urinary retention were reported, while the anastomotic leakage and pelvic collection did not differ. Overall, the length of stay was significantly longer by 2 days for one-team vs. two-team patients. The surgical outcomes are summarized in Table 2. 

The prognostic factors were entered into a multivariate analysis to assess the risk of an anastomotic leak. An American Society of Anesthesiologist physical status of 3 and higher and a long operation time were independent predictors of an anastomotic leak, while obesity, sex, age, anastomosis height, and the surgical approach (one-team vs. two-team) were not, as presented in Table 3.

### 3.3. Histologic Outcomes

Total mesorectal excision achieved a consistent high quality, with 160 (97%) specimens graded complete or near complete. Similarly, 96% of the resection margins were clear, with a median distal resection margin of 1.6cm (IQR 1–3) and a median circumferential margin of 10mm (IQR 5–15). 

Three-quarters of the patients underwent neoadjuvant radiochemotherapy, which translates to 31/124 patients (25%) who had a pathologic complete response. Overall, about half of the tumors were stage pT3 (44%), and 39 (24%) patients harbored tumor-positive lymph nodes, whereas a median of 27 lymph nodes (IQR 20–38) were harvested. The Dvorak tumor regression grade showed a major response (TRG 3 and 4) in 37.6% of the patients. Table 4 summarizes the histologic results.

### 3.4. Oncologic Outcomes

Median follow-up was 50 months (range 14 to 118 months, IQR 32–79 months). 

A unifocal local recurrence occurred in nine patients (5.4%), and no patient suffered a multifocal recurrence. The local recurrences were diagnosed at a median of 13 months (IQR 12–36, range 6–45 months). The distal and circumferential resection margins were shorter, albeit not significantly, in patients suffering from a local recurrence (8.5 mm (IQR 2.5–20) vs. 17.5mm (IQR 10–30), *p* = 0.089; 7 mm (IQR 3.5–11.5) vs. 10mm (IQR 5–15), *p* = 0.279, respectively). Patients who recurred locally were prone to bear a distant recurrence (62.5% vs. 27%, *p* = 0.031), and their survival was reduced due to the local and distant recurrences (*p* < 0.001). Six out of nine patients with a local recurrence presented with a distant metastatic disease, while three had only a locoregional failure. Remarkably, a local recurrence was also diagnosed in a late follow up at 36, 44, and 45 months. 

Of the six patients who had an R1 resection, three had a metastatic disease within 4 months of a taTME (at diagnosis twice and at 4 months once), and a single patient recurred locally 12 months after a taTME, whereas no distant metastatic disease was seen until currently (at a 22-month follow up). In the multivariable analysis, a positive distal resection margin was independently associated with a local recurrence, as presented in Table 5.

A distant metastatic disease was present in 44 patients (26.7%), including 17 patients (10.3%) who were operated on with a known limited liver metastatic disease and who underwent either a primary or secondary curative liver metastatis resection. There were no differences in sex, BMI, tumor height, one-team vs. two-team approaches, and circumferential resection margin between the cohorts of patients who bore a distant metastatic disease and those who did not. Conversely, the presence of tumor-positive lymph nodes was associated with a distant metastatic disease in the univariable analysis and independently in the multivariable analysis, as shown in Table 6.

The entire cohort reached a median DFS of 36 months (IQR 18–66) and an OS of 46 months (IQR 29–78). The survival analysis showed a DFS of 82% at 1 year, 78% at 2 years, 73% at 3 years, 67% at 4 years, and 67% at 5 years. OS was 95% at 1 year, 92% at 2 years, 91% at 3 years, 91% at 4 years, and 90% at 5 years. Survival was not influenced by the one-team vs. two team surgical approaches.

### 3.5. Resource Utilization

A taTME was performed using the one-team (n = 55) and two-team (n = 110) techniques, depending on staff availability. The median operation time for the one-team approach was 422 min (IQR 353–492), significantly longer than for the two-team approach, which had a median surgery time of 320 min (IQR 276–373, *p* < 0.001). Indeed, opting for the two-team approach saved 102 min or 24% of the theatre time and reduced the length of the hospital stay by 2 days, while the anastomotic leak rate and oncological outcomes remained similar. Since the Swiss reimbursement system is based on a diagnosis-related group compensation system, shortening the hospital stay by 2 days has a limited influence on the direct financial return for the hospital. Conversely, freeing up an in-patient bed earlier has a noticeable impact on clinical care because of chronic hospital bed and staff shortages.

In terms of resource utilization, a two-team approach required the engagement of a second surgeon and a second scrub nurse for less than 3 h. The additional cost of the second scrub nurse and of a second piece of laparoscopy equipment was accounted for in the internal pricing of theatre use, while the additional cost of a second surgeon was largely represented by the opportunity cost in a salary-based compensation model. In this prospective cohort, a two-team approach saved EUR 1384.75 per procedure, while freeing up enough theatre time to perform another minor procedure in the same operating room on the same day. This was indeed observed with in-patient procedures of up to a two-hour duration routinely being listed ahead of a two-team taTME, whereas a one-team taTME was rarely preceded by any significant procedure other than minor outpatient surgery.

## 4. Discussion

The present prospective study investigated the surgical and oncologic outcomes of a taTME, in the context of worrisome data from Norway and the Netherlands that reported a multifocal local recurrence in up to 10% of patients [32] and an anastomotic leakage in up to 25% of patients [33]; both these figures are twice as high as can be reasonably expected. Consequently, Norway and the United Kingdom have enforced a taTME moratorium, and many national regulatory bodies are scrutinizing the outcomes. Importantly, the worrisome reports were representative of the early learning curve, as most surgeons had performed fewer than 10 solo taTMEs. The two taTME surgeons who operated on all patients in the present cohort had performed many hundreds of laparoscopic TMEs and undergone taTME cadaveric courses before embarking in clinical taTME practice. The surgical quality was high, with no intraoperative complications other than three (1.8%) CO_2_ embolisms, which did not prevent the safe completion of any taTMEs and just a single conversion (0.6%) to an open TME. Thirty-day major postoperative morbidity (Clavien–Dindo ≥ 3b, requiring surgical, endoscopic, or radiological intervention under general anesthesia) affected 13 patients, including 3 deaths (1.2%) and 12 (7.2%) anastomotic leakages, which could all be salvaged endoscopically, whereas the median length of stay was 9 days, and the time to an ileostomy closure was 2 months. In the multivariable analysis, preoperative frailty (ASA grade ≥ 3) and a long operation time predicted anastomotic leakage. These figures compare well with those of the latest taTME cohort from the international taTME registry [20] and with the most recently published comparative data and meta-analysis between open, laparoscopic, robotic, and transanal TMEs [10,34,35,36]. 

The surgical oncologic quality was likewise high: 3% with an incomplete mesorectum and 3.6% with resection margins harboring a microscopically residual tumor (96.4% R0 resection); almost half of the patients had a bulky T3 cancer, and 75% underwent neoadjuvant radiochemotherapy. A complete pathologic response ypT0 was achieved in 25% of the 124 patients who received long-course radiochemotherapy. Staging pelvic MRI revealed 42 threatened circumferential margins, 76% ultimately being cleared by neoadjuvant treatment and a taTME. The median distal resection margin on the formalin fixed specimen was 1.5 cm (IQR 10–28 mm), with 0.5–1 cm of anastomotic ring to be added. Hence, a cancer-tailored TME was performed, and an unduly generous distal margin was avoided for the sake of optimal function. Finally, the large number of retrieved lymph nodes, with a median of 27 in patients who mostly underwent neoadjuvant treatment, confirmed the high surgical oncology quality. 

The observed rate of local recurrence was 5.4% (9/165), and 26.7% of the patients suffered from distant metastatic diseases after a median follow-up of 50 months (IQR 32–79 months). All local recurrences were unifocal, and they occurred at a median of 13 months after the taTME. Of the patients who recurred locally, 62.5% presented with a distant metastatic disease during a further follow-up. A positive distal margin was an independent predictor of a local recurrence in the multivariate analysis, underlining one of the unique benefits of a taTME. Indeed, visualization of the distal tumor edge when performing a taTME allows for tailoring the distal resection margin to the individual patient, hence taking into account both the oncologic and functional perspectives. Conversely, in randomized trials, a robotic TME was associated with the longest distal resection margin when compared to open, laparoscopic, and transanal TMEs, with no oncologic benefit, though there was the potential for worse functional outcomes [9,10]. The oncologic outcomes of the present prospective cohort are consistent with a recently published large multi-centred cohort series [37]. Further large cohort studies of patients who received a taTME, with a follow-up of 2 years and more, reported a local recurrence rate between 3% [24] and 7.4% [38]. The International taTME Registry reported 2803 patients who were followed for 2 years with a local recurrence rate of 4.8% and 9.6% unifocal pelvic recurrences that were detected at a median time of 14 months for a 2-year DFS and OS of 77% and 92%, respectively. Moreover, the AlaCaRT and ACOSOG randomized trials showed local recurrences in 4.6% and 5.4% of patients who underwent laparoscopic TME [6,7], respectively, even though patients with a threatened margin during preoperative MRI were excluded from AlaCaRT, as opposed to the present prospective cohort of unselected consecutive patients. Finally, 7% of AlaCaRT and 12% of ACOSOG patients had an R1 resection, whereas in the Norwegian report that triggered a national taTME moratorium, every fifth patient with a local recurrence had an R1 resection [32]. Accordingly, the present cohort series compares well to the literature, with a unifocal local recurrence rate of 5.6%, an R1 rate of 3.6%, and a 5-year DFS and OS of 67% and 90%, respectively. The present analysis, hence, concurs with the growing body of literature establishing the high oncologic quality that is achievable using a taTME approach, with a high-quality resection being consistently achieved in >96% of unselected patients presenting with a distal rectum cancer, when performed by specialized surgeons.

In terms of resource utilization, a taTME is more expensive than a conventional laparoscopic TME, while requiring much less upfront investment and a lower cost per case than a robotic approach [39]. A two-team taTME maximized the benefit of a transanal approach, with superior visualization of the critical structures and easier dissection of the most challenging part of the TME, plus it sped up the theatre time and led to an effective TME with mutual support between the abdominal and transanal surgeons. Still, additional manpower resources are required when a two-team approach is opted for. Taking advantage of a second scrub nurse and surgeon generates its own costs, while decreasing the surgery time by 102 min, or 24% of the whole procedure time. In economic terms, saving 102 min of theatre time reduces the total cost of a taTME by EUR 1385, even when accounting for the additional manpower. Moreover, a two-team approach frees up close to 2 h of theatre time when compared to a one-team approach, allowing for the scheduling of another case in the same operating room, e.g., a laparoscopic hernia repair or a cholecystectomy—which may translate to a further efficiency gain and cost savings. Last but not least, in the setting of a teaching hospital, performing a two-team taTME doubled the safe teaching opportunities for the residents and fellows. Indeed, participation in taTME cases was much more appreciated by the surgeons in training than scrubbing into a robotic case as a table assistant.

Practicing a taTME safely requires proper training and mentoring and a proper surgical case load, as advised in the consensus guidance [22,23]. The learning curve of a taTME was estimated to be 30–70 cases long [40,41], with a minimum institutional load of 30 TMEs being recommended [23], while the best surgical and oncological outcomes were reported in institutions that performed 50 or more TMEs yearly [42,43]. Nationwide structured training programs in the UK and the Netherlands have recently been implemented, including proper mentoring and proctoring [33,44]. Such programs are an invaluable asset when implementing a taTME, together with the inclusion/benchmarking of own progress in a prospective clinical registry. The present prospective cohort series as well as nationwide reports from Denmark, Canada, and the International taTME Registry, totaling many thousands of patients operated on in expert centers, report an anastomotic leak rate below 10% and a pelvic recurrence rate lower than 5% [20,21,45,46], emphasizing the importance of proper training and adequate clinical exposure before embarking on a solo taTME practice. In terms of surgical complications, prehabilitation can improve physical status, which was independently associated with an anastomotic leak in the present cohort, while specialization and centralization can reduce the procedure time and surgical morbidity [43,47,48]. Overall, the present real-world data contribute to establishing the safety and benefits of a taTME in a teaching hospital with no patient selection and residents and fellows actively participating in the cases under direct supervision.

## 5. Conclusions

A taTME is a technically demanding procedure owing to an unusual anatomical perspective and peculiar pitfalls that are unknown in a classical TME. However, the benefits of a taTME in terms of precise margin control, a high-quality TME dissection, and single stapling are obvious, in particular when dealing with a bulky tumor within a fatty, irradiated mesorectum in a narrow pelvic. The present prospective cohort series, of 165 patients with a long-term follow up, confirms the safety, reproducibility, and oncological high quality achieved with a taTME, in contrast to the worrisome figures reported from Norway and the Netherlands. It also provides economic support for a two-team approach and the maximization of the surgical, oncologic, and educational benefits of a taTME, which has results that are at least as good as those using the current robotic platform, with comparatively low investment and procedure costs.

## Figures and Tables

**Table 1 cancers-15-01190-t001:** Demographics and pretreatment tumor characteristics.

Variable	Overall (*n* = 165)	One-Team (*n* = 55)	Two-Team (*n* = 110)	*p-*Value
Gender, n (%)				0.816
Female	55 (33)	19 (35)	36 (33)	
Male	110 (67)	36 (65)	74 (67)	
Median age, years (IQR)	64 (55–76)	69 (58–76)	62 (53–73)	0.030
Median body mass index, kg/m^2^ (IQR)	26 (23–29)	26 (23–29)	26 (22–29)	0.946
ASA risk classification, n (%)				0.481
ASA 1–2	88 (53)	32 (58)	56 (51)	
ASA 3–4	77 (47)	23 (42)	54 (49)	
Median tumor distance to anal verge by rigid proctoscopy, cm (IQR)	7 (5–10)	7 (5–9)	7 (5–10)	0.749
Preoperative circumferential resection margin in MRI, n (%)				0.450
Positive	42 (25)	16 (22)	26 (24)	
Negative	123 (75)	39 (78)	84 (76)	
Neoadjuvant radio-chemotherapy, n (%)				0.204
Yes	124 (75)	38 (69)	86 (78)	
No	41 (25)	17 (31)	24 (22)	

ASA, American Society of Anesthesiology score; IQR, interquartile range; MRI, magnetic resonance imaging.

**Table 2 cancers-15-01190-t002:** Surgical outcomes.

Variable	Overall (*n* = 165)	One-Team (*n* = 55)	Two-Team (*n* = 110)	*p-*Value
Median operative time, minutes (IQR)	348 (293–425)	422 (353–492)	320 (276–373)	<0.001
Median height of anastomosis from anal verge, cm (IQR)	6 (4–8)	6 (4–8)	6 (4–8)	0.715
Rate of conversion, n (%)	1 (1)	0	1 (1)	0.480
Highest complication grade at 90 days, Clavien–Dindo grading (%)				0.030
0	74 (45)	21 (38)	53 (48)	
1	23 (14)	3 (5)	20 (18)	
2	32 (19)	14 (26)	18 (16)	
3a	22 (13)	11 (20)	11 (10)	
3b	7 (4)	3 (5)	4 (4)	
4a	4 (2)	2 (4)	2 (2)	
4b	0	0	0	
5	3 (2)	1 (2)	2 (2)	
Most frequent complications, n (%)				
Anastomotic leakage	12 (7)	5 (9)	7 (6)	0.526
Pelvic collection	7 (4)	1 (2)	6 (6)	0.276
Bleeding	3 (2)	3 (6)	0	0.014
Urinary retention	18 (11)	10 (18)	8 (7)	0.035
Paralytic ileus	23 (14)	9 (16)	14 (13)	0.526
LOS, days (IQR)	9 (7–13)	11 (9–14)	9 (7–11)	0.002
Mortality at 30 days, n (%)	2 (1)	0	2 (2)	0.316
Reoperation rate at 30 days, n (%)	13 (8)	5 (9)	8 (7)	0.684
Mortality at 90 days, n (%)	3 (2)	1 (2)	2 (2)	1.000
Reoperation rate at 90 days, n (%)	14 (9)	6 (11)	8 (7)	0.410
Reversal of ileostomy, n (%)	142 (86)	46 (84)	96 (87)	0.727

IQR, interquartile range; LOS, length of stay.

**Table 3 cancers-15-01190-t003:** Multivariate analysis of anastomotic leakage in patients who underwent taTME.

Variable	OR (95% CI)	*p-*Value
Male sex	5.589 (0.6–49.0)	0.120
High ASA grade (≥3)	5.655 (1.1–29.2)	0.039
Age ≥ 65 years	1.463 (0.3–6.3)	0.608
Obesity (body mass index ≥ 30 kg/m^2^)	0.409 (0.1–2.3)	0.310
Neoadjuvant chemotherapy	0.387 (0.1–1.6)	0.192
Low anastomosis (≤6 cm from anal verge)	0.298 (0.1–1.3)	0.104
Two-team approach	1.387 (0.3–6.6)	0.682
Long operation time (>75th percentile)	7.045 (1.2–41.6)	0.031

**Table 4 cancers-15-01190-t004:** Histologic outcomes.

Variable	Overall (*n* = 165)	One-Team (*n* = 55)	Two-Team (*n* = 110)	*p-*Value
Mesorectal resection quality in Quirke grade, n (%)				0.788
Grade 1, incomplete	5 (3)	0	5 (5)	
Grade 2, nearly complete	11 (7)	6 (11)	5 (5)	
Grade 3, complete	149 (90)	49 (89)	100 (90)	
T-stage, n (%)				0.289
0	31 (19)	9 (16)	22 (20)	
1	18 (11)	4 (7)	14 (13)	
2	43 (26)	15 (28)	28 (25)	
3	73 (44)	27 (49)	46 (42)	
4	0	0	0	
N-stage, n (%)				0.723
Negative	126 (76)	41 (75)	85 (77)	
Positive	39 (24)	14 (25)	25 (23)	
Lymph nodes harvested, median (IQR)	27 (20–38)	26 (20–38)	27 (20–38)	0.864
cM-stage, n (%)				0.737
0	145 (88)	49 (89)	96 (87)	
1	20 (12)	6 (11)	14 (13)	
L-stage, n (%)				0.358
0	155 (94)	53 (96)	102 (93)	
1	10 (6)	2 (4)	8 (7)	
V-stage, n (%)				0.518
0	139 (84)	45 (82)	94 (86)	
1	26 (16)	10 (18)	16 (14)	
Pn-stage, n (%)				0.865
0	152 (92)	51 (93)	101 (92)	
1	13 (8)	4 (7)	9 (8)	
Resection margin, n (%)				1.000
Negative	159 (96)	53 (96)	106 (96)	
Positive	6 (4)	2 (4)	4 (4)	
Distal resection margin, mm (IQR)	16 (10–30)	15 (9–25)	18 (10–30)	0.383
Circumferential resection margin, mm (IQR)	10 (5–15)	9 (4–15)	9 (5–16)	0.417
Dvorak tumor regression grade, n (%)				0.663
1	32 (19)	12 (22)	21 (19)	
2	71 (43)	21 (38)	49 (45)	
3	21 (13)	9 (16)	12 (11)	
4	41 (25)	13 (24)	28 (25)	

**Table 5 cancers-15-01190-t005:** Multivariate analysis of local recurrence.

Variable	OR (95% CI)	*p-*Value
Male sex	0.768 (0.1–4.8)	0.778
Obesity (body mass index (≥30 kg/m^2^)	0.244 (<0.1–3.2)	0.283
Low anastomosis (≤6 cm from anal verge)	0.548 (0.1–3.7)	0.537
High T-stage (≥3)	1.571 (0.2–10.5)	0.641
Positive N-stage	0.829 (0.1–5.5)	0.846
Two-team approach	1.785 (0.2–13.5)	0.574
Long operation time (>75th percentile)	6.003 (0.6–62.3)	0.133
Positive distal resection margin	29.565 (2.1–418.1)	0.012
Positive circumferential margin	3.283 (0.3–33.5)	0.316
Anastomotic leakage	0.908 (0.1–15.6)	0.947

**Table 6 cancers-15-01190-t006:** Multivariate analysis of distant recurrence.

Variable	OR (95% CI)	*p-*Value
Male sex	1.982 (0.7–5.3)	0.174
Obesity (body mass index (≥30 kg/m^2^)	0.434 (0.1–1.5)	0.188
Low anastomosis (≤6 cm from anal verge)	1.432 (0.6–3.7)	0.454
High T-stage (≥3)	1.270 (0.5–3.2)	0.614
Positive N-stage	3.189 (1.3–7.9)	0.012
Two-team approach	0.490 (0.2–1.4)	0.169
Long operation time (>75th percentile)	0.366 (0.1–1.4)	0.144
Positive distal resection margin	0.737 (0.1–6.7)	0.787
Positive circumferential margin	2.855 (0.6–14.4)	0.204
Anastomotic leakage	2.664 (0.6–11.6)	0.192

## Data Availability

The data presented in this study are available on request from the corresponding author.

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
