# Peer review of "Surgical Outcomes, Long-Term Recurrence Rate, and Resource Utilization in a Prospective Cohort of 165 Patients Treated by Transanal Total Mesorectal Excision for Distal Rectal Cancer"

_cancers, 2023, doi:10.3390/cancers15041190_

Round 1
Reviewer 1 Report
The authors publish the data from their prospective monocentric cohort study on taTME. The methodology is very well presented and the data are excellently processed. The presentation is comprehensive and includes all relevant parameters. The very good results show all relevant problems of deep rectal resections and the authors describe the critical complications in detail. The cost analysis under Swiss billing conditions is also relevant as a new aspect.
Introduction and discussion are very extensive, but also illuminate almost all aspects of the topic. A very good and interesting publication.
Additions/explanations are relevant and desirable on two points:
-What is the authors' rationale for setting the indication for taTME at 10 cm? They describe > 50 rectal resections in their own clinic. What other resections were these besides taTME? If the tumour height is 10 cm, then only PME? If from 10-12 cm a TME was also performed, why were not all TMEs operated with the taTME technique?
-The database is consecutive and prospective. How long was the learning curve of the main surgeons before data collection and are the current very good results worse in the learning curve?
Author Response
Reviewer 1
The authors publish the data from their prospective monocentric cohort study on taTME. The methodology is very well presented and the data are excellently processed. The presentation is comprehensive and includes all relevant parameters. The very good results show all relevant problems of deep rectal resections and the authors describe the critical complications in detail. The cost analysis under Swiss billing conditions is also relevant as a new aspect.
Introduction and discussion are very extensive, but also illuminate almost all aspects of the topic. A very good and interesting publication.
Additions/explanations are relevant and desirable on two points:
-What is the authors' rationale for setting the indication for taTME at 10 cm? They describe > 50 rectal resections in their own clinic. What other resections were these besides taTME? If the tumour height is 10 cm, then only PME? If from 10-12 cm a TME was also performed, why were not all TMEs operated with the taTME technique?
-The database is consecutive and prospective. How long was the learning curve of the main surgeons before data collection and are the current very good results worse in the learning curve?
We thank this Reviewer for the kind appraisal of our work and for the opportunity to clarify both points raised.
- Proximal rectal cancers (above 10cm from the anal verge) were routinely resected laparoscopically, including partial mesorectal excision, thus avoiding unnecessary total mesorectal excision which was only performed in distal rectal cancer, in agreement with current consensus recommendation and guidelines. This point was marked in green in the revised manuscript.
- TaTME training and implementation occurred in 2013, including cadaver course and selection of cases. The present series started in 2014 under the leadership of 2 experts surgeons and inclusion of all/consecutive cases into the prospective study database and into the International taTME registry. Over time, operative efficiency increased and technique matured as we learned from our own practice and by exchanging with other taTME experts. Thankfully, there were no statistically significant changes in complication rates and oncologic outcomes between the first 20 cases and the last 20 cases. Most likely, the good outcomes reported from the early days are a tribute to the large prior experience of the 2 surgeons involved, who had performed many hundreds TME before starting taTME. The learning curve of taTME has been estimated to be 30-70 cases long: the present data suggest that a large prior experience in laparoscopic TME and performing taTME by two expert TME surgeons in a structured setting allowed safe implementation and optimized outcomes. These points were highlighted in green in the revised manuscript.
Reviewer 2 Report
The topic is of interest to colorectal surgeons and the manuscript is basically well structured and well written.
There are some issues which the authors need to consider.
General considerations
The manuscript is generally too long, especially the introduction (nearly 700 words) and the discussion parts (>1500 words), but all other sections as well. The manuscript would clearly benefit from a substabtial reduction of the text mass.
There is too much repetition of previously presented results, often in detail, in the discussion part.
It is good that the authors discuss the very discouraging results from notably Norway but also to some extent the Netherlands. It is obvious to me that the surgical teams in these countries, especially Norway, had not prepared well enough to undertake a new and complex procedure such as taTME for rectal cancer surgery which is seldom easy regardless of technique. The authors show a good example of sound surgical preparations, high volume for a limited number of surgeons, and present very acceptable short- and long term results. It would be of interest if the authors discussed how they view the future of conventional laparoscopy, and possibly even robotic surgery, in an institution which has as good results with taTME as in their institution, and is there in fact even a future place for the robot at their department?
Specific considerations
What is actually meant by "with a two-team approach being the default option" - not quite obvious to the present reviewer.
The authors need to adress the difference in age between the one-team and the two-team. Do they think there was any systematic selection or a difference by chance.
The authors also need to comment the significant difference in time in hospital between the one- and two-team approaches. Do the authors find that this difference has a rational explanation or could it be by pure chance. Please expand.
Presenting survival curves for DFS and OS with the comparison between the one- and two-team approaches is of limited interest considering the unclear mechanism preceeding this selection, and therefore these two survival curves may be omitted.
It is noteworthy that as much as 75% of the patients had neoadjuvant CRT, as well as the substancial proportion of cCR. Would seem like promising conditions for the watch and wait concept. Please comment and expand.
Author Response
Reviewer 2
The topic is of interest to colorectal surgeons and the manuscript is basically well structured and well written. There are some issues which the authors need to consider.
General considerations
The manuscript is generally too long, especially the introduction (nearly 700 words) and the discussion parts (>1500 words), but all other sections as well. The manuscript would clearly benefit from a substabtial reduction of the text mass.
There is too much repetition of previously presented results, often in detail, in the discussion part.
It is good that the authors discuss the very discouraging results from notably Norway but also to some extent the Netherlands. It is obvious to me that the surgical teams in these countries, especially Norway, had not prepared well enough to undertake a new and complex procedure such as taTME for rectal cancer surgery which is seldom easy regardless of technique. The authors show a good example of sound surgical preparations, high volume for a limited number of surgeons, and present very acceptable short- and long term results. It would be of interest if the authors discussed how they view the future of conventional laparoscopy, and possibly even robotic surgery, in an institution which has as good results with taTME as in their institution, and is there in fact even a future place for the robot at their department?
We thank this Reviewer for the critical appraisal of our work. We have shortened the manuscript and avoided redundancy, while providing the additional information requested. Concerning the role of robotic surgery in rectal cancer, in line with the present results and the conclusion of this study, we do not see any clinical benefit for a robotic approach for low rectal cancer, as commented in the discussion section (text marked in blue). Ongoing RCT have a subgroup comparing taTME to robotic TME with preliminary results expected within 2 years.
Specific considerations
What is actually meant by "with a two-team approach being the default option" - not quite obvious to the present reviewer.
We thank this Reviewer for the opportunity to clarify this: two-team taTME was our standard whenever possible, with staff availability and OR list constraints leading to one-team taTME in many occasions. The rationale for a two-team approach is highlighted in blue. We now write "with a two-team approach being the preferred option" in the revised manuscript.
The authors need to adress the difference in age between the one-team and the two-team. Do they think there was any systematic selection or a difference by chance.
The statistically significant age difference of 7 years between one-team and two-team patients occurred by chance and was not the product of a deliberate selection. Age had no influence in uni- and multivariate analysis neither on septic complications including anastomotic leakage, nor on oncologic recurrences.
The authors also need to comment the significant difference in time in hospital between the one- and two-team approaches. Do the authors find that this difference has a rational explanation or could it be by pure chance. Please expand.
We believe that the reduction of LOS of stay between one-team and two-team approaches may be partially explained by a reduction in surgery time and trauma. Yet, this view is purely speculative, as many factors are know to influence LOS and the data at hand do not allow to draw a firm conclusion in a study with a nearly 10-year duration. Importantly, neither a one-team nor a two-team approach were associated with a change in morbidity or in oncologic outcomes in uni- and multivariable analysis. This point is stressed in the revised manuscript.
Presenting survival curves for DFS and OS with the comparison between the one- and two-team approaches is of limited interest considering the unclear mechanism preceeding this selection, and therefore these two survival curves may be omitted.
We were pleased to remove both figures as requested.
It is noteworthy that as much as 75% of the patients had neoadjuvant CRT, as well as the substancial proportion of cCR. Would seem like promising conditions for the watch and wait concept. Please comment and expand.
The present series investigated patients with distal rectal cancer, in whom neoadjuvant radiochemotherapy is very often recommended according to current treatment guidelines of ESMO, which the interdisciplinary tumor board adhered to. Similarly, watch and wait was not recommended as this approach is not broadly validated but experimental and mandates inclusion in a dedicated clinical trial. The rate of 25&% of complete response to neoadjuvant radiochemotherapy is similar to what other authors have published. Overall, adherence to international treatment guidelines is mandatory in our centre and audited by national certifiying authorities, as detailed in the revised manuscript (text marked in blue).